

# Machine learning prediction of the mass and the velocity of controlled single-block rockfalls from the seismic waves they generate

Clément Hibert[1], Francois Noël[2,3], David Toe[5], Miloud Talib[1], Mathilde Desrues[1], Emmanuel Wyser[2], Ombeline Brenguier[6], Franck Bourrier[4], Renaud Toussaint[1,7], Jean-Philippe Malet[1], and Michel Jaboyedoff[2]

[1]Institut Terre et Environnement de Strasbourg / ITES, CNRS & University of Strasbourg F-67084 Strasbourg, France
[2]Institut des Sciences de la Terre / ISTE, University of Lausanne, Géopolis, CH-1015 Lausanne, Switzerland
[3]Geological Survey of Norway, NO-7491 Trondheim, Norway
[4]Université Grenoble Alpes, INRAE, ETNA, 38000 Grenoble, France
[5]Université Grenoble Alpes, INRAE, LESSEM, 38000 Grenoble, France
[6]Société Alpine de Géotechnique / SAGE, FR-38160 Gières, France
[7]SFF Porelab, The Njord Centre, Department of Physics, University of Oslo, P. O. Box 1048, Blindern, N-0316 Oslo, Norway

**Correspondence:** Clément HIBERT (hibert@unistra.fr)

**Abstract.** Understanding the dynamics of slope instabilities is critical to mitigate the associated hazards but their direct observation is often difficult due to their remote locations and their spontaneous nature. Seismology allows us to get unique information on these events, including on their dynamics. However, the link between the properties of these events (mass and kinematics) and the seismic signals generated are still poorly understood. We conducted a controlled rockfall experiment in

5 the Riou-Bourdoux torrent (south French Alps) to try to better decipher those links. We deployed a dense seismic network and inferred the dynamics of the block from the reconstruction of the 3D trajectory from terrestrial and airborne high-resolution stereo-photogrammetry. We propose a new approach based on machine learning to predict the mass and the velocity of each block. Our results show that we can predict those quantities with average errors of approximately $10\%$ for the velocity and $25\%$ for the mass. These accuracies are as good as or better than those obtained by other approaches, but our approach has

10 the advantage of not requiring to localize the source and an *a priori* knowledge of the environment, nor of making a strong assumption on the seismic wave attenuation model. Finally, the machine learning approach allows us to explore more widely the correlations between the features of the seismic signal generated by the rockfalls and their physical properties, and might eventually lead to better constrain the physical models in the future.

## 1 Introduction

15 Slope instabilities are complex natural phenomena that pose a threat to humans and infrastructures in many regions of the World. Landslides, rockfalls, rock avalanches and surface collapses generating pit craters are natural disasters that can affect our societies. They also play a major role in the Earth surface dynamics as important erosion processes, whose occurrence might be caused by external factors such as Earthquakes, intense precipitation or the thawing of ice in the joints and fractures



of large rocky masses for example. Understanding the triggering mechanisms, their dynamics, quantifying and documenting
their properties and their spatio-temporal occurrences is of paramount importance to mitigate the associated risks but also to
understand their contributions to long and short-term erosion processes. However, because of their spontaneous and destructive
nature, gravitational instabilities are difficult to study.

Over the past two decades, these processes have been increasingly studied through the use of approaches based on seismology. Seismology makes it possible to complete the source of information conventionally deployed to study mass wasting
processes (e.g. direct testimony, remote sensing, geomorphology, geodetic measurements, etc.) by its ability to provide information on event properties such as its exact time of occurrence (to the seconds) and its localisation (e.g., Norris, 1994; Deparis
et al., 2008; Yamada et al., 2012; Hibert et al., 2014a; Dammeier et al., 2011, 2016; Gracchi et al., 2017; Dietze et al., 2017;
Allstadt et al., 2018; Yan et al., 2019; Kuehnert et al., 2020b), with the possibility of recording them over vast distances (up to
1000 kilometers for the largest events) (e.g., Kanamori and Given, 1982; Kanamori et al., 1984; Ekström and Stark, 2013; Allstadt, 2013; Hibert et al., 2019). More than providing spatio-temporal information, sometimes in real-time, seismology offers
the possibility to retrieve the dynamics of an event through the information carried by the seismic signal emitted during the
triggering and the propagation of the event. There are very few other observational approaches that allow retrieval of important
insights on the dynamics. Hence finding relationships between seismic signals generated by gravitational instabilities and their
properties has been a major focus of recent research in landslide and rockfall seismology.

For catastrophic landslides (volume over 1 million cubic meter), approaches based on the inversion of the long-period (lowfrequency, below 0.5 Hz) seismic waves have been proposed. By retrieving the force exerted by the mass displacement on the
Earth those approaches have successfully helped to determine dynamic parameters (velocity, momentum, acceleration) and
properties of these events (e.g., Kawakatsu, 1989; Ekström and Stark, 2013; Allstadt, 2013; Zhao et al., 2012; Iverson et al.,
2015; Hibert et al., 2014b, 2017a; Moore et al., 2017; Dufresne et al., 2019; Li et al., 2017; Moretti et al., 2020; Chao et al.,
2018; Zhang et al., 2019). However most of mass wasting processes that occur worldwide do not have a volume large enough
to generate those long-period waves, thus precluding the use of inversion methods to retrieve their dynamics quantities. Yet,
those mass wasting processes will most of the time generate high-frequency seismic waves (frequency above 1 Hz). Being able
to infer physical properties from those high-frequency seismic waves will therefore allow us to characterize most mass wasting
processes, with either small or catastrophic volumes, which is critical to have a better understanding of the occurrence and the
physics of those phenomena and thus for mitigating the risks they generate.

Recent studies proposed scaling laws between high-frequency seismic signal features and source properties of rockfalls
and landslides. These studies are mostly based on laboratory experiments (e.g., Farin et al., 2015, 2016, 2019; Arran et al.,
2020), real-scale experiments (e.g., Bottelin et al., 2014; Hibert et al., 2017b; Saló et al., 2018), and documented natural events
(e.g., Norris, 1994; Deparis et al., 2008; Dammeier et al., 2011; Hibert et al., 2011; Levy et al., 2015; Hibert et al., 2017a;
Le Roy et al., 2019). Among the quantities studied, several correlations between the mass and the velocity of the rockfall, and
the magnitude, the maximum amplitude at the source and the seismic energy of the seismic signal have been observed and
sometimes quantified. Several scaling laws have been proposed (e.g., Norris, 1994; Deparis et al., 2008; Hibert et al., 2011;
Levy et al., 2015; Hibert et al., 2017b; Saló et al., 2018; Le Roy et al., 2019) but are all carrying strong uncertainties, caused



mainly by the simplicity of the propagation models used (e.g., Le Roy et al., 2019; Kuehnert et al., 2020a), the difference

of contexts (soft soil vs. hard rock, influence of the seismic network geometry) and sometimes even of the physics of the

source (free-fall, granular flows, single rockfall, multiple rockfalls). However all those studies demonstrated that there is a

link between some seismic signals features (maximum amplitude at the source, seismic energy, local magnitude) and some

source properties (mass, velocity, energies, momentum, force or acceleration). The difficulty resides now in understanding

the fundamental physics that explains those correlations, as well as in increasing the accuracy of the scaling laws proposed.

This is deemed important as it opens the perspective to quantify mass movement dynamics directly from the seismic signals

they generate (i.e. without inversion or modelling). This is critical for the development of future methods aimed at their real-

time detection and characterization using high-frequency seismic signals. This can be achieved by improving both the source

physical model and seismic waves propagation model which remains a strong challenge. These improvements require more

high-quality observations to calibrate and validate the models. This is what motivated the 2018 Riou-Bourdoux controlled

rockfall experiment, which followed and improved upon a similar experiment conducted in 2015 (Hibert et al., 2017b).

Thanks to the deployment of a dense seismological network close to the block impacts, and an approach allowing an accurate

reconstruction of the trajectories (Noël et al., 2022), we tried to complete three objectives: 1) Better understand and model the

propagation of the seismic waves generated by the block impacts; 2) Find and try to better constrain the correlations between

the kinematic parameters of the impacts of the blocks and the features of their seismic signals; 3) Explore the use of an

innovative approach based on a machine learning algorithm to infer the mass and the velocity of the block at each impact from

the seismic signals they generate.

## 2    Material and methods

### 2.1    Context : the Riou-Bourdoux catchment

The Riou-Bourdoux is a torrential catchment located in the South French Alps, approximately 4 km north of the city of

Barcelonnette (France). It formed in callovo-oxfordian black marls whose high erosion susceptibility resulted in the formation

of numerous steep (> 30 degrees) gullies. The blocks were launched in a gully located on the north slope of the torrent. The

travel path had a length of approximately 200 m and slope angles ranging from 45 degrees on the upper part of the slope to

approximately 20 degrees on the terminal debris cone (Figure 1). The launched elements consisted in hard limestone blocks

selected in the torrent and brought to the launch pad with a backhoe.

### 2.2    Block trajectories and properties measurements

Kinematic parameters of each launch were computed from 31 reconstructed rockfall trajectories using the ballistic equations

of a free-falling object neglecting the drag from the air (Volkwein et al., 2011; Wyllie, 2014; Loew et al., 2021). The back cal-

culation method using 3D terrain models and video footage (Noël et al., 2017; Noël et al., 2022) requires accurately measuring



the geometric features of each launched block and of the terrain, and to track their propagation with high speed multispectral
cameras from different view angles.

31 limestone blocks were individually weighted using a lift and a tension load cell. The density of the rocks were determined
in the laboratory from analysis conducted on core samples taken from each block. The block shapes were acquired using mobile
handheld terrestrial laser scans (mobile terrestrial laser scanning / GeoSLAM ZEB-Revo) and from Structure-from-Motion
photogrammetry (SfM) using pictures acquired with a Panasonic GH5 camera and the software Agisoft Metashape Pro v.1.4.4.
The LiDAR model has a spatial density of about 50,000 points per $m^2$ at the block level. The SfM model was build from 128
photos for each block and has a density of about 5 millions points per $m^2$ when scaled (average: $4.93 10^6$ pts/$m^2$; standard
deviation: $2.123 10^6$ pts/$m^2$). Assuming a homogeneous distribution of the mass, the moments of inertia of each block and the
main axes of inertia were identified from the 3D models of each block and the density. Their dimensions were measured on the
3D models aligned on their main axes of inertia.

A very-high resolution terrain model of the gully (Figure 1) was acquired using four acquisition methods to ensure proper
coverage of occluded faces, detailed texture of the surfaces and accurate scale and orientation relative to the horizontal. A
highly detailed terrain SfM model was generated from georeferenced pictures acquired with a DJI Phantom 4 UAV flying at
an average altitude of 25.3m. We use the software Agisoft Metashape Pro v.1.4.4. The model was built from 167 photos, with
resolution of 5472×3078 pixels, and with a selected overlap of at least 9 images. The initial model had 345922467 points, with
a ground resolution of 6.32mm/pix, and was downscale to 83475710 points spaced by 1 cm. Its scale was then adjusted by less
than 1% using the iterative closest point algorithm to match with a detailed terrain model obtained from four locations with a
terrestrial laser scanning device (Optech ILRIS-LR). The main gully was also scanned with a mobile terrestrial laser scanning
while rappelling down, to cover every part in detail. Finally, evenly spread targets were painted in the upper and lower part of
the gully and were located using a laser theodolite.

The blocks were pushed down manually one by one separated by about 5 to 10 minutes. Their trajectories were manually
tracked from up to 8 viewpoints: five viewpoints had fixed framing, being installed on tripods (one in the middle part of the
travel path and four at the bottom of the gully); two viewpoints were from the sky using two DJI drones, one flying in hover
and one following the motion of the blocks; the last viewpoint was from a camera panned manually to track the rocks using a
long-focus lens and was located at the bottom of the gully.

## 110   2.3   Seismic network and data

The seismic network was deployed along the gully. The network comprised 16 3-components geophones (4.5Hz/3C connected
to a Daqlink seismic camera at a sampling rate of 1000 Hz). The exact position of the sensors were measured by differential
GNSS (Figure 1). In this analysis, we used only the vertical components of the geophones as we observed the best signal-
to-noise ratio on this component. Data from the geophones number 14 and 16 were discarded as the records exhibited high
amplitude noises and spikes probably related to a faulty connection or a bad installation. Before analysis each record was
deconvolved from the instrument response to get the ground velocity. No filtering was applied to the raw data.



## 2.4 Trajectory and kinematics reconstruction

The impact locations of each block were pointed on the 3D textured detailed terrain model (Figure 1). The task was eased by using a custom developed software (Noël et al., 2022) in which the terrain can be visualised from the same viewpoints as the corresponding video footage, and in which the reconstructed trajectories offset by the radius of each rock are updated in real-time following the cursor mouse or manually entered impact coordinates. The position and time of each impact can thus be accurately defined until obtaining visually matching trajectories with those visible in the camera footage. With non-optimal viewing angles or terrain texture with little contrast, screenshots of the terrain model and video footage were aligned with the Handle Transform Tool in the GIMP software using the surrounding elements of texture in order to find the exact location of the impact.

The trajectories were exported with their velocities and vectors normal to the terrain and the center of mass of the blocks is extrapolated from the impact position on the ground. All trajectories were further visually inspected in the CloudCompare software. The angular velocities were obtained by averaging the number of block revolutions performed during the period in between each impact. The average axis around which the block rotated was identified to estimate the angular momentum based on the geometric features of each block.

In total, 376 impacts were available from 25 trajectory segments composed of many free-falling parabolas. The impacts at the extremities of each segment are missing because of missing incoming/outcoming velocities. Therefore, 326 impacts were reconstructed with their incoming and outgoing translational and angular velocities, kinetic energy changes and momentum.

## 2.5 Trajectories and seismic records synchronization

While the seismological data could be time-stamped by a GNSS, the clocks of the different cameras used during the experiment are not all set to the absolute time. To determine the lag between the two times series (time of impact from the direct observations and seismic records) with a precision below the second, we performed a cross-correlation analysis. The timing of the impacts was transformed into a time series of zeros and ones, zeros indicating the times with no impact and ones the time of each impact. We then normalized the seismic records by the maximum of the envelope and computed the cross-correlation between the impacts time series and the normalized envelope of the seismic records, with lags ranging from minus 10 seconds to plus 10 seconds. The lag for which the best normalized correlation was observed was selected. A manual control and final adjustment of the results has been performed. After this first step we manually picked the beginning and the end of each seismic signal on each station. We selected only the signals associated with impacts that did not result in the fracturing of the blocks and that were not generated by parts of fragmented blocks. This was verified for each impact on the videos of the launches. We also selected only impacts for which it was possible to pick clearly the beginning and the end of the seismic signal and therefore discarded all intricate and low amplitude seismic signals. An example of the seismic signals recorded at one station and of the selected impact seismic signals is presented in Figure 2. This resulted in a dataset of 384 seismic signals of impacts.



## 2.6 Seismic sources parameters computation

There are essentially two properties of the high-frequency seismic signals generated by mass movements that have been studied
in correlation with the physical parameters of the source dynamics, the maximum amplitude of the seismic signal corrected for propagation effects $A_0$, and the energy of the seismic signal at the source $E_s$ (e.g., Norris, 1994; Deparis et al., 2008; Dammeier et al., 2011; Schneider et al., 2011; Hibert et al., 2011; Bottelin et al., 2014; Levy et al., 2015; Farin et al., 2015, 2016; Hibert et al., 2017b, a; Saló et al., 2018; Le Roy et al., 2019; Farin et al., 2019; Arran et al., 2020). These two quantities are usually compared to the source velocity, momentum, and its kinetic and potential energies. Both quantities are computed from
attenuation parameters that allow to account for the attenuation of seismic waves caused by the propagation of waves in the Earth and which are caused by geometrical spreading and anelastic attenuation. Determining an adequate attenuation model is therefore critical.

Thanks to the reconstruction of the trajectories, in our study we know the exact location of the impact and hence the exact distances between the source and the receivers, thus we could test several attenuation models and find the one that
better explains the observed decay of the amplitudes with the distance. The best model should be the one that allows the best regression of the maximum amplitude of each impact recorded at each station as a function of the distance of those stations to the location of the impact.

We tested three simple attenuation models, one for surface wave (Eq. 1) and one for body wave (2), both proposed by Aki and Chouet (1975), which consider the anelastic attenuation of seismic waves through the use of the attenuation factor $\beta$, as
well as the possibility of a simple linear attenuation of the maximum amplitude with distance (Eq. 3).

$$A(r) = A_0 \frac{e^{-\beta r}}{\sqrt{r}}, \tag{1}$$

$$pA(r) = A_0 \frac{e^{-\beta r}}{r}, \tag{2}$$

$$A(r) = \frac{A_0}{r}, \tag{3}$$

The maximum amplitude at the source $A_0$ and the $\beta$ factor can be determined directly from the attenuation model for each
impact.

An approximation of the seismic energy for body-waves can be computed as Crampin (1965):

$$E_s = \int_{t_i}^{t_f} 4\pi r^2 \rho c u_{env}(t)^2 e^{\beta r} dt, \tag{4}$$



with :

$$u_{env}(t) = \sqrt{u(t)^2 + Ht(u(t))^2}, \qquad (5)$$

where $Ht$ is the Hilbert transform of the seismic signal $u(t)$ used to compute the envelope $u_{env}(t)$, $t_i$ and $t_f$ the times of the beginning and the end of the seismic signal respectively and $\rho$ the density of the layer through which the generated surface waves propagate, and $c$ their phase velocity. The average velocity of body waves in black marls is approximately 450 m.s$^{-1}$ (Hibert et al., 2012; Gance et al., 2012). The density $\rho$ of dry black marls is approximately 1450 kg m$^{-3}$ (Maquaire et al., 2003). For each impact we computed the seismic energy at each station and kept the mean over all stations.

**2.7   Machine Learning: using Random Forests as a regression tool**

Random Forests (Breiman, 2001) is a machine learning algorithm based on the computation of a large number of decision trees. Decision trees are top-down structures consisting of nodes and branches. At each node a statistical test is performed on the value of one feature of the input data. The outcome of this test tells which branch to use to get the next node. The final nodes of the tree give the decision of the tree. The randomness comes from the use of a random subset of events from the 185 dataset and of features used to characterize the events to build each tree. Each decision tree in the "forest" is therefore different and the model combines hundreds (if not thousands) of decision trees.

Random Forests is now successfully used in seismology for automated source classification (Provost et al., 2017; Hibert et al., 2017c; Maggi et al., 2017; Malfante et al., 2018; Hibert et al., 2019; Ao et al., 2019; Pérez et al., 2020; Wenner et al., 2021; Chmiel et al., 2021). However the Random Forests algorithm can also be used to estimate continuous values and thus 190 perform regression analyses. The model will then not give a class (e.g. an integer) but an estimation of a value that exists in a continuum. In other words, a Random Forests classifier is able to identify the origin of a seismic source (for example landslides, earthquakes, mining blasts, etc.) while a Random Forests regressor is able to "predict" (in a statistical machine learning sense) the time of occurrence of laboratory-triggered earthquakes (e.g., Rouet-Leduc et al., 2017). For a classification application of the Random Forests algorithm the predicted class is given by the majority vote of all the trees. For a regression, the mean of 195 the predicted values by each tree is the final result.

In this study we exploit Random Forests as a regression tool to "predict" the mass and the velocity of the rockfalls from the features of the seismic signal generated by each impact at the ground. The methodology consisted in: 1) defining relevant seismological features to characterize the data; 2) defining a subset of the dataset to train the Random Forests model; 3) training the model and 4) testing the model on a subset of the dataset (the test set) not selected for the training. To assess the robustness 200 and estimate uncertainties associated, steps 2 to 4 are repeated hundreds of times, by increasing, from 10 to 100, the number of events in the training set.

When selecting seismic signals features we must find those that might carry the most relevant information on the source properties. We choose 57 features proposed by Provost et al. (2017) and Hibert et al. (2017c) and given in Appendix A. Those features are used for many applications of the Random Forests as an automated seismic source classifier. They can be



categorized into three families: 1) waveform features (temporal); 2) spectral (frequency) features and 3) pseudo-spectrogram (evolution of the frequency content with time) features. When analysing a dataset from multiple stations, it might be complicated to merge the information carried by all signals in the same set of features. Thus, the strategy chosen consisted in computing the feature on each station for a given impact and taking the mean of a given feature value on all stations. We also computed the standard deviation of the values of each feature over all the stations and included those standard deviations in

our array of features. Hence we have a set of 114 features for each impact (57 mean values and 57 standard deviation values) and we consider each impact seismic signal as a sample in the dataset. As for the $A_0$ and $E_S$ computation, we considered only the impact for which the attenuation regression model yields a determination coefficient above $0.6$. The maximum amplitude at the source $A_0$ and the seismic energy $E_S$ are not included in the features used.

By analysing the machine learning model produced we can determine which features of the seismic signals carry the im-

portant information that the model is using to successfully predict the value of the mass and the velocity of the block at each impact. This might provide insights on the link between the dynamics of the block and the seismic source. This is possible by computing the importance score of each feature, which accounts for the relative contribution of each feature in the success of the regression. The value of the importance of each feature is computed by permuting the values of a given feature in the features array, and assessing how this permutation impacts the regression results. If the permutation of a given feature value

results in a worse overall fitting of the real values than the predicted ones, then the feature is important in the regression process. Conversely, if the prediction accuracy remains the same while permuting a feature value, then this feature has little impact in the regression process. The importance is given by a normalized score. The higher is the score of the feature the higher is its importance in the prediction process.

In this work we set the number of decision trees in the forest to 1000. We trained and tested the machine learning model

with an increasing number of samples, from 10 to 100 with a step of 10. For each case (10 to 100 samples), we repeated the process of training and testing the algorithm 100 times, to assess the robustness of the model.

## 3    Results

### 3.1    Attenuation models

Figure 3 shows the maximum amplitude recorded at each station for each impact of the launch of Block #1. The maximum

amplitude of the signal is decreasing with the distance $r$ of the sensor to the location of the impact as expected. For each attenuation model we computed the regression line and assess the quality of the regression by computing the determination coefficient $R^2$. This was performed for each selected impact. The mean of the $R^2$ coefficient for the body wave model, the surface wave model and the geometrical attenuation model are, $0.70$, $0.64$ and $0.48$ respectively. For 363 over a total of 384 impacts, the best regression model between the maximum amplitude and the distance between the impact and the sensors is

the model 2, which assumes body-waves propagation. Therefore for the computation of $A_0$ and $E_S$ we choose to use the body wave model. For the analysis of the correlation and the test of the machine learning approach we selected the 298 impacts for which the attenuation model was able to fit the real data with a coefficient $R^2$ of at least $0.6$. All the other impacts were



excluded to avoid including too peculiar events. Low $R^2$ values might be explained by irregular kinematic behaviours such as the block hitting an obstacle (trees, other rocks), multi-impacts in a very short time, composite contacts or sliding of the block, 240 or an impact being too far from the seismic network.

## 3.2  Correlations between the seismic and trajectography parameters

For 298 impacts we analyse the relationship between two seismic quantities ($A_0$ and $E_S$) and nine kinematic parameters : the incident northbound, eastbound and vertical velocity and the incident velocity modulus ($Vix$, $Viy$, $Viz$ and $|Vi|$), the incident and the rebound momentum (Pi and Pf), the incident and rebound kinetic energy ($Ei$ and $Ef$) and the difference between those 245 two energies ($Ef - Ei$). The X-axis is oriented east to west and the Y-axis is oriented south to north. For each pair, we tested simple linear regressions and computed determination coefficients (Figure 4).

The best correlations are observed between the incident velocity modulus $|Vi|$ and the maximum amplitude at the source $A_0$, and between the incident kinetic energy $E_i$ and the seismic energy $E_s$, with determination coefficient $R^2$ of 0.43 and 0.39 respectively. The worst correlation is observed between the northbound velocity and $A_0$ with a $R^2$ of 0.04.

## 3.3  Mass and velocity predictions

We assessed the quality of the predicted results by computing the difference in percent between the predicted and the real values of the mass and of the modulus of the velocity. Therefore a difference of $0\%$ is reached when the predicted value is equal to the real value. In table 1 we present the median error of the prediction on the 100 instances of training-test the algorithm as a function of the number of samples used to train the model (10 to 100). The median values, which are less impacted by outlier 255 values, are reported in Tab 1. The mean, the median and the complete distribution of the error on the prediction of the mass and the velocity for the cases of model training with 10 to 100 samples are presented on Figure 5.

| Number of training samples | Average error on velocity [%] | Average error on mass [%] |
|---|---|---|
| 10 | 19.0 | 43.3 |
| 20 | 16.3 | 39.0 |
| 30 | 15.2 | 36.6 |
| 40 | 13.9 | 34.6 |
| 50 | 13.4 | 32.9 |
| 60 | 12.7 | 31.1 |
| 70 | 12.1 | 29.8 |
| 80 | 11.6 | 28.6 |
| 90 | 11.2 | 26.6 |
| 100 | 10.7 | 25.3 |

**Table 1.** Predictions result: percentage of error between the real and the predicted values





As shown in Table 1 and Figure 5, with 10 samples used to train the model, we reach a median of the prediction error of 43.3% on the mass and 19.0% on the velocity. Those values drop to 32.9% and 13.4% for 50 samples, and to 25.3% and 10.7% for 100 samples.

## 3.4 Features importance

Figure 6 presents the mean importance scores of the features for models aiming at predicting the mass and the velocity and trained with 100 samples. For the mass prediction, the 20 best features are based on the waveforms (8 features) and the pseudo-spectrograms (11 features). Only one spectral feature appears in the top-20. The 5 most important features are the mean of the seismic energy in the 5-10 Hz frequency band (#13), the mean of the seismic energy in the 10-30 Hz frequency band (#14), the mean ratio between the envelope of the maximum frequency over the envelope of the mean frequency (#43), the mean ratio between the envelope of the second quartile of the frequency spectrum over the envelope of the first quartile of the frequency spectrum (#55), and the mean ratio between the envelope of the third quartile of the frequency spectrum over the envelope of the first quartile of the frequency spectrum (#57).

For the velocity prediction, the 20 best features are also mostly based on the waveforms (10 features) and the pseudo-spectrograms (7 features), with only three spectral features appearing in the top-20. The 5 most important features are the standard deviation of the seismic energy in the 100-200 Hz frequency band (#74), the mean of the seismic energy in the 100-200 Hz frequency band (#17), the standard deviation of the values of the energy of the seismic signal in the 50-100 Hz frequency band (#72), the standard deviation of the difference between the envelope of the maximum frequency over the envelope of the median frequency (#111) and the standard deviation of the values of the energy of the seismic signal in the 30-50 Hz frequency band (#71).

We can note that 1) none of the best 5 features are the same for the mass and the velocity prediction, 2) only 6 features are common in the top-20 for both quantities and 3) mass prediction uses none of the features computed from the standard deviation of the features computed at each station (features with numbers above #57) while the model for velocity prediction uses 4 of them in the top-5. Finally most of the top 5 features for the mass and the velocity prediction are based on a difference of energy in several frequency bands.

## 4 Discussion

### 4.1 Correlations between the seismic and trajectography parameters

Figure 4 shows qualitative correlations between the momentum, the kinetic energy, the maximum amplitude at the source and the seismic energy, as observed or modelled in previous studies (Deparis et al., 2008; Vilajosana et al., 2008; Hibert et al., 2011; Levy et al., 2015; Farin et al., 2015; Hibert et al., 2017b; Farin et al., 2016; Saló et al., 2018; Le Roy et al., 2019). Our results suggest that the kinetic energy before impact is better correlated to the seismic energy than the loss of kinetic energy between the impact and the rebound $Ef - Ei$. The block travel directions were mostly from West to East along the gully morphology.





The lack of strong displacement in the North-South direction, and hence the low velocity values, might explain the poorest correlation observed between $Viy$ and $A_0$.

However most $R^2$ values are low for all the correlations investigated. Those weak quantitative correlations precluded us from using the scaling laws to estimate the mass and the velocity of the blocks at each impact as proposed in (Hibert et al., 2017b) because it would result in very high uncertainties on the inferred masses and velocities. As demonstrated by (Kuehnert et al., 2020a), velocity-depth profile, 3-D soil heterogeneities, source direction and the topography play a major role in the modulation of the waveforms and the amplification of both the maximum amplitude and the energy of the generated seismic

signals. Those effects are not taken into account in the simple attenuation models used in this study and numerous previous ones. We are starting to have access to complex models that can take into account some of these effects for high frequency seismic signals (Kuehnert et al., 2020a), but they require high computational time and a comprehensive knowledge of the context physical properties (velocity profile, 3-D medium heterogeneities, etc.), which can be difficult to get for real conditions. Having access to these models to perform direct modeling or inversion of the source parameters might be laborious and expensive to

reproduce in different contexts, preventing an hypothetical easy portability of the approach for operational uses. This motivated the exploration of the machine learning approach to infer the properties of the rockfall without needing any attenuation model or an $apriori$ knowledge of the medium.

## 4.2   Seismic signal features importance and physical model

The force imparted by an elastic sphere on a solid elastic surface can be described by the Hertz contact theory (Hertz, 1882),

as proposed by (Farin et al., 2015), and was demonstrated to be relevant to model the force created by a block impacting the ground in experimental and natural experiments (Farin et al., 2015; Bachelet et al., 2018; Kuehnert et al., 2020a). These studies have shown that, in the framework provided by the Hertz theory, the seismic signals maximum amplitude, energy, corner frequency or spectrum width are controlled by the velocity, the mass, the duration of the impact and the physics and the geometry of the contact of a single block with the ground. Therefore the seismic signals maximum amplitude, energy, corner

frequency or spectrum width carry information on the dynamics and properties of the impacting block, and might be analysed to retrieve those physical quantities, and especially the force, the velocity and the mass of the impactor.

    The Random Forest model we trained yields information on which features of the seismic signal carry the most important information to successfully predict the mass and the velocity. We observe that the most important features used to predict the velocity are not exactly the same as those used to predict the mass. However the absolute seismic energy in several frequency

bands (Features #13-17 and #70-74) is an important information for both the prediction of the mass and of the velocity. This is consistent with the work by (Farin et al., 2015), which have shown that the radiated seismic energy and the frequency content of a seismic signal generated by an individual impactor scales with its mass and velocity. Hence by including the energy of the seismic signal filtered in different frequency bands as features in our predictive model we can retrieve this correlation and allow the model to do accurate prediction.

On this specific set of features (#13-17 and #70-74) we observe a discrepancy between the importance of the features used for the prediction of the mass and those used for the prediction of the velocity. The standard deviation of the feature values



(hence the difference between the values of the feature at each station) is impactful for the prediction of the velocity but not for the prediction of the mass. This suggests that the difference of seismic energies recorded at different stations is an important information for the prediction of the velocity (not for the mass). Moreover the energy in the lower frequency bands are important features for the prediction of the mass, while it is the energy in the highest frequency band that is an important feature for the prediction of the velocity (which is also visible on features 37 and 94). Those observations suggest an effect of the propagation of the waves, a filtering process, that helps to determine the true values of the velocity. This might suggest that the feature of the seismic signals recorded at the closer stations are more important for the determination of the velocity, as the high frequency are the first attenuated by the propagation of the seismic waves. The detail of the process and understanding why it only affects the velocity prediction are however difficult to unravel from our dataset and should be more thoroughly investigated, through laboratory experiments for example. This observation is not inconsistent with the Hertz theory.

Regarding the frequency content, according to the feature importance the full spectrum (FFT) of the whole signal carries less information than the spectrograms and the filtered waveforms. This is unexpected as according to the Hertz theory the full spectrum of the signal (maximum amplitude, width, corner frequency) should all be highly dependent on the mass and the velocity of the impactor. This suggests that the temporal variation of the seismic signal spectrum (i.e. spectrograms) is more important in the prediction process and hence carry more information on the source properties than the information we can obtain from the full frequency spectrum itself.

Finally we are more accurately predicting the velocity of the block at the impact than its mass with the 114 selected features. According to Kuehnert et al. (2020a) who predicted the impact forces for two impacts or real rockfalls occurring at the Piton de la Fournaise Volcano using the Hertz contact theory, the maximum impact force is highly sensitive to the variation of the impact speed whereas the frequency content is the most sensitive to the density of the impactor and the Young modulus of the impactor and the impacted plane. In our case all the blocks and impacted zones had similar elastic properties. This may suggest that the variability of the impacted forces imparted by our block on the ground was controlled essentially by a change of the velocity, the change of the mass playing a less important role. This might translate into the velocity having a higher control on the information carried by the seismic signal and thus being more easily predictable by the machine learning model, and might explain some of the observations made previously.

## 5 Conclusions and perspectives

From the experimental single-block controlled launches conducted in the Riou-Bourdoux torrent, we demonstrated that a machine learning model based on the Random Forest algorithm is able to provide estimate of the mass and the velocity of the block at each impacts with an average error of around 25% for the mass and 10% for the velocity. With this new approach, we obtain a prediction accuracy on these two quantities equivalent to or better than all previous studies focusing on the high frequencies of the seismic signals generated by mass movements, which gave errors ranging from 20% to 400% of the target values (e.g., Hibert et al., 2011; Dammeier et al., 2011; Farin et al., 2015; Hibert et al., 2017b; Le Roy et al., 2019).





The machine learning model uses solely the features of the recorded signals and does not require an attenuation model to
estimate the source properties conversely to the approaches based on the computation of the seismic energy and the maximum
amplitude at the source. This removes the need to make assumptions which are necessary in the classical approaches used until
now but which are carrying strong uncertainties, such as the velocity of the seismic waves, the density of the soil, the anelastic
attenuation factor and the attenuation model used. The machine learning approach also removes the need to know the exact
localisation of the impacts and to correct for site effects. Those are major advantages for an operational implementation of such
methods for rockfall risks assessment and mitigation. An implementation in any context will only require to perform several,
well-monitored, controlled launches of rockfalls to produce a dataset to train the machine learning model, which will then be
able to predict the mass and the velocity of future rockfalls. Another strength of the Random Forest approach is its ability to
perform well even with few events used to train the algorithm. Finally we use the same seismic signal features to predict the
mass and the velocity of rockfalls that are already used to detect and identify seismic sources associated with mass wasting
processes (Provost et al., 2017; Hibert et al., 2017c; Maggi et al., 2017; Wenner et al., 2021). This opens the prospect to build a
detection system, based on seismic waves, that is able to tell when a rockfall occurs, what is its mass and velocity and possibly
its localisation, all at the same time and even in near real-time given the possibility to easily record and broadcast seismic data.

It is further important to note that this experiment was performed in a controlled context, with an ideal setup, with simple
mono-block rockfalls which travelled roughly along the same path, and with a seismic network very close to the sources.
The machine learning based approach must now be experienced with more complex sources, such as multi-blocks rockfalls
and even granular flows, and with more distant seismic stations. The station distances might hinder the ability of the machine
learning model to estimate source properties, as the farthest we are from the source, the more we lose information due to
propagation effects on seismic waves. However, the recent successes (Provost et al., 2017; Hibert et al., 2019; Wenner et al.,
2021; Chmiel et al., 2021) in identifying mass wasting sources at medium to long distances, with the same approach and the
same features, suggest that even when recording seismic signals far from the source, seismic signals retain information on the
source properties in the higher frequency band (above 1 Hz), that could allow to determine those properties using the same
approach. This would be a major breakthrough as it would allow to determine source properties for most landslides which do
not generate seismic waves with enough energy in the lowest frequency bands to allow for an inversion of the properties of the
source. This will be the subject of future work.

Finally, this approach based on machine learning algorithms might be applied to the analysis of other environmental pro-
cesses for which classical seismological source inversion methods are not suitable. This could be used for the determination of
properties (mass, velocity, flux, volume, forces, momentum, etc.) of sources that generate tremors (volcanic eruptions, debris
flows, intense storms), complex high-frequency and even low-frequency signals (ice-calving events, hydro-acoustic signals) or
even anthropogenic noises (vehicles, pumps). However, as for every machine learning based approach, sets of calibrated and
well known examples are necessary to train the models. Physical models can also help by producing physically-based synthetic
seismic signals. Regression of seismic source properties using machine learning approaches is a new complementary and inter-
esting tool for the community interested in exotic or environmental seismic sources relevant for improving our understanding
of these processes.



*Code and data availability.* All the pre-processed data, the raw seismic data and the code to compute the signal features are accessible at
https://doi.org/10.5281/zenodo.6393210.

*Author contributions.* C.H., F.N., M.J., J.-.P.M. and F.B. conceptualized the research. C.H. processed the seismic data, implemented the machine learning approach and wrote the original draft. F.N., D.T. and F.B. processed the trajectory data. F.N., M.J., A.C., C.H., F.B and J.P.M. validated the trajectory reconstruction approach. All the authors participated and helped to conduct the controlled launch experiment. All the authors reviewed and edited the original draft. J.-.P.M. and M.J. funded this research.

*Competing interests.* The authors declare no competing interests

*Acknowledgements.* This work was carried with the support of the French National Research Agency (ANR) through the projects HY-DROSLIDE "Hydrogeophysical Monitoring of Clayey Landslides", the Open Partial Agreement "Major Hazards" of Council of Europe through the project "Development of Cost-effective Ground-based and Remote Monitoring Systems for Detecting Landslide Initiation", the Research Council of Norway through its Centres of Excellence funding scheme, Project No. 262644. and the Observatoire Multi-disciplinaire
des Instabilités de Versant (OMIV) (RESIF/OMIV, 2015). The authors thank H. Collomb (ONF-RTM / Alpes-de-Haute-Provence) for facilitating the access to the Riou-Bourdoux experimental site and Pierre Bottelin for insightful comments on the manuscript.



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





**Figure 1.** Orthophotography of the Riou-Bourdoux gully, with the reconstructed trajectory of all the blocks, and the location of the geophones used in this study indicated by colored dots. The color of the trajectory scale with the absolute translational velocity of the block. The raw seismic signals recorded at each geophone for the first launch are represented on the right, in the color corresponding to the one of the dots indicating the location of the sensor.





**Figure 2.** Seismic signal (a) and spectrogram (b) generated by impacts of the Block #1 and recorded on geophone 1. The selected seismic signals used in our analysis are indicated in blue.





**Figure 3.** Maximum envelope amplitude as a function of the distance for each impact and each geophone for the block #1. The colour corresponds to the colour of the geophones on Figure 1. The black line indicates the best regression computed with the model assuming a simple linear attenuation of the maximum amplitude with distance (3), the light grey one assuming a signal dominated by surface waves (1) and the dark gray one assuming a signal dominated by body waves (2).







**Figure 4.** Correlation between the seismic and trajectography properties of the blocks: a) the eastbound incident velocity, b) the northbound incident velocity, c) the vertical incident velocity, d) the modulus of the incident velocity and e) the incident total momentum and f) the restituted momentum as a function of the maximum amplitude at the source $A_0$; g) the incident kinetic energy, h) the restituted kinetic energy and i) the difference of both as a function of the seismic energy $E_s$. The black line indicates the best linear regression, with the coefficient of determination $R^2$ indicated in the panel. Dots of the same color are from the same rockfall launch (i.e. identical block mass).





**Figure 5.** Distribution of the error (%) over 100 instance of training and testing the Random Forest model for the prediction of the mass values when trained with 10 to 100 samples to predict a) the velocity and b) the mass. The mean error is indicated by a black line, the median by a red line.



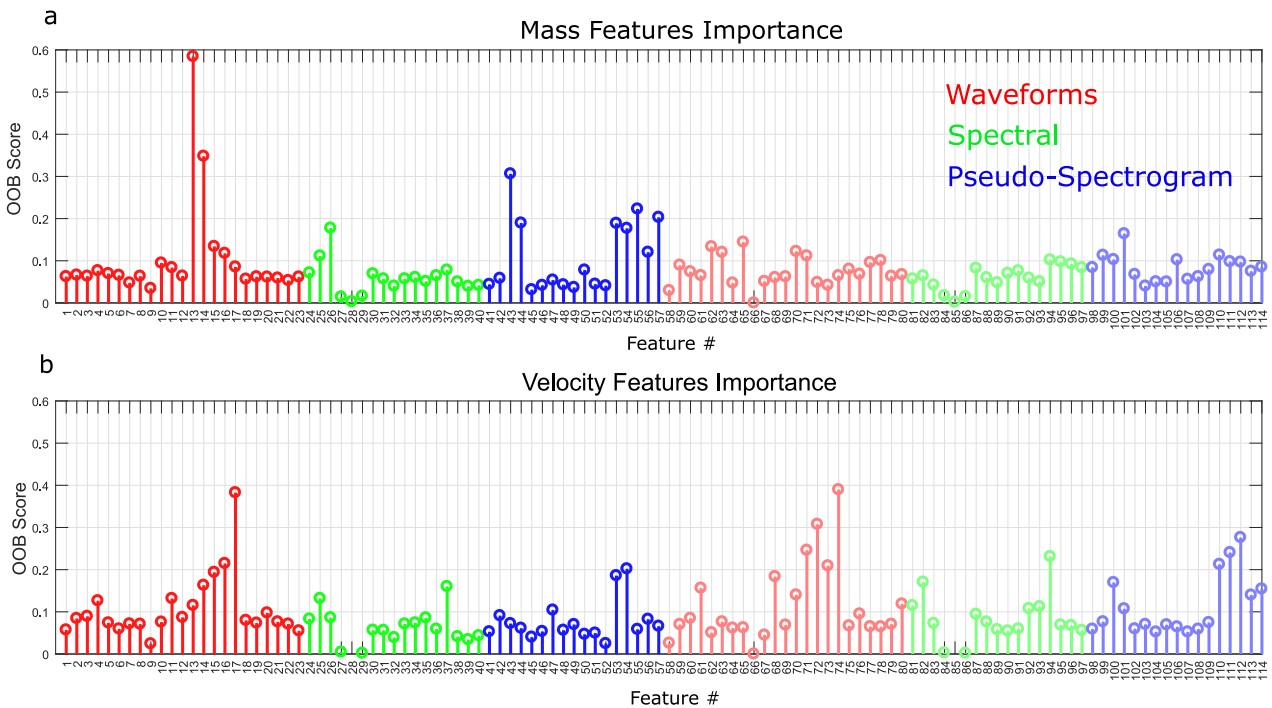

**Figure 6.** Importance score of the features for the prediction of a) the mass and b) the velocity. Colors indicate the family of features (waveform, spectral or pseudo-spectrogram). The description of each feature and their respective numbers can be found in Appendix A.



**Table A1.** Features table

| Number [mean(std)] | Name | Formula |
|---|---|---|
| | *Waveform attributes:* | |
| 1 (58) | Duration | $T = t_e - t_s$ |
| 2 (59) | RappMaxMean | $\max[e(t)]/\mathrm{mean}[e(t)]$ |
| 3 (60) | RappMaxMedian | $\max[e(t)]/\mathrm{median}[e(t)]$ |
| 4 (61) | AsDec | $(t_{\max} - t_s)/(t_e - t_{\max})$ |
| 5 (62) | KurtoSig | $\mathrm{Kurt}[s(t)]$ |
| 6 (63) | KurtoEnv | $\mathrm{Kurt}[e(t)]$ |
| 7 (64) | SkewSig | $\mathrm{Skew}[s(t)]$ |
| 8 (65) | SkewEnv | $\mathrm{Skew}[e(t)]$ |
| 9 (66) | CorPeakNumber | number of peaks in $a(\tau)$ |
| 10 (67) | Energy1/3Cor | $\int_0^{\mathrm{T}/3} \mathrm{a}(\tau)\mathrm{d}\tau$ |
| 11 (68) | Energy2/3Cor | $\int_{\mathrm{T}/3}^{\mathrm{T}} \mathrm{a}(\tau)\mathrm{d}\tau$ |
| 12 (69) | int_ratio | $\int_0^{\mathrm{T}/3} \mathrm{a}(\tau)\mathrm{d}\tau / \int_{\mathrm{T}/3}^{\mathrm{T}} \mathrm{a}(\tau)\mathrm{d}\tau$ |
| 13-17 (70-74) | ES1 to ES5 | $\mathrm{ES}_i = \log_{10} \int e_i(t) dt$ |
| 18-22 (75-79) | KurtoF1 to KurtoF5 | $\dfrac{\mathrm{Kurt}[s_i(t)]}{}$ |
| 23 (80) | RMSDecPhaseLine | $\sqrt{\overline{e(t) - l(t)}^2}$ |
| | *Spectral attributes:* | |
| 24 (81) | MeanFFT | $\mathrm{mean}[|S(\nu)|]$ |
| 25 (82) | MaxFFT | $\max[|S(\nu)|]$ |
| 26 (83) | FMaxFFT | $\nu_{\max}$ |
| 27 (84) | MedianFFT | $\mathrm{median}[|S(\nu)|]$ |
| 28 (85) | VarFFT | $\mathrm{var}[|S(\nu)|]$ |
| 29 (86) | FCentroid | $\mathrm{centroid}[|S(\nu)|]$ |
| 30 (87) | Fquart1 | $\mathrm{centroid}[|S(\nu)|_1]$ |
| 31 (88) | Fquart3 | $\mathrm{centroid}[|S(\nu)|_3]$ |
| 32 (89) | NPeakFFT | number of peaks in $|S(\nu)| > 0.75|S(\nu)|_{\max}$ |
| 33 (90) | MeanPeaksFFT | $\mathrm{mean}[|S(\nu)| \mathrm{at\ peaks}]$ |
| 34-37 (91-94) | E1FFT to E4FFT | $\mathrm{E}_i \mathrm{FFT} = \int |S(\nu)|_i d\nu$ |
| 38 (95) | gamma1 | $\gamma_1 = \sum \nu|S(\nu)|^2 / \sum|S(\nu)|^2$ |
| 39 (96) | gamma2 | $\gamma_2 = \sqrt{\sum \nu^2|S(\nu)|^2 / \sum|S(\nu)|^2}$ |
| 40 (97) | gammas | $\sqrt{|\gamma_1^2 - \gamma_2^2|}$ |
| | *Pseudo-spectrogram attributes:* | |
| 41 (98) | KurtoMaxDFT | $\mathrm{Kurt}[\max[|\mathrm{DFT}(t,\omega)|]]$ |
| 42 (99) | KurtoMedianDFT | $\mathrm{Kurt}[\mathrm{median}[|\mathrm{DFT}(t,\omega)|]]$ |
| 43 (100) | MaxOverMeanDFT | $\mathrm{mean}[\frac{\max[|\mathrm{DFT}(t,\omega)|]}{\mathrm{mean}[|\mathrm{DFT}(t,\omega)|]}]$ |
| 44 (101) | MaxOverMedianDFT | $\mathrm{mean}[\frac{\max[|\mathrm{DFT}(t,\omega)|]}{\mathrm{median}[|\mathrm{DFT}(t,\omega)|]}]$ |
| 45 (102) | NbrPeaksMaxDFT | Number of peaks in $\max[|\mathrm{DFT}(t,\omega)|]$ |
| 46 (103) | NbrPeaksMeanDFT | Number of peaks in $\mathrm{mean}[|\mathrm{DFT}(t,\omega)|]$ |
| 47 (104) | NbrPeaksMedianDFT | Number of peaks in $\mathrm{median}[|\mathrm{DFT}(t,\omega)|]$ |
| 48 (105) | Ratio between 45 and 46 | $-$ |
| 49 (106) | Ratio between 45 and 47 | $-$ |
| 50 (107) | NbrPeaksCentralFreq | Number of peaks in $\mathrm{median}[|\mathrm{DFT}(t,\omega_2)|]$ |
| 51 (108) | NbrPeaksMaxFreq | Number of peaks in $\mathrm{median}[|\mathrm{DFT}(t,\omega_{\max})|]$ |
| 52 (109) | Ratio between 50 and 51 | $-$ |
| 53 (110) | DistMaxMeanFreqDTF | $\mathrm{mean}[\max[|\mathrm{DFT}(t,\omega)|] - \mathrm{mean}[|\mathrm{DFT}(t,\omega)|]]$ |
| 54 (111) | DistMaxMedianFreqDTF | $\mathrm{mean}[\max[|\mathrm{DFT}(t,\omega)|] - \mathrm{median}[|\mathrm{DFT}(t,\omega)|]]$ |
| 55 (112) | DistQ2Q1DFT | $\mathrm{mean}[\mathrm{centroid}[|\mathrm{DFT}(t,\omega)|_2] - \mathrm{centroid}[|\mathrm{DFT}(t,\omega)|_1]]$ |
| 56 (113) | DistQ3Q2DFT | $\mathrm{mean}[\mathrm{centroid}[|\mathrm{DFT}(t,\omega)|_3] - \mathrm{centroid}[|\mathrm{DFT}(t,\omega)|_2]]$ |
| 57 (114) | DistQ3Q1DFT | $\mathrm{mean}[\mathrm{centroid}[|\mathrm{DFT}(t,\omega)|_3] - \mathrm{centroid}[|\mathrm{DFT}(t,\omega)|_1]]$ |

Waveform- and spectrum-based features, with $s(t)$ the windowed raw seismogram, $e(t)$ its envelope,
$l(t) = e_{max} - \frac{e_{max}}{t_f - t_{max}}t$, $a(\tau)$ its auto-correlation function, $s_i(t)$ the windowed seismograms filtered in the 5–10
Hz ($i = 1$), 10–30 Hz ($i = 2$), 30–50 Hz ($i = 3$), 50–100 Hz ($i = 4$), 100–199 Hz ($i = 5$) frequency bands, $e_i(t)$ their
corresponding envelopes, $t_s$ and $t_e$ the start and end times of the window, $t_{max}$ the time of the maximum amplitude,
$\mathrm{Kurt}(X) = \frac{\mu_4(X)}{\sigma^4(X)}$ the Kurtosis of distribution $X$ where $\mu_4(X)$ indicates the fourth moment of $X$ and $\sigma$ its
standard deviation, $\mathrm{Skew}(X) = \frac{\mu_3(X)}{\sigma^3(X)}$ the Skewness of distribution $X$ where $\mu_3$ indicates the third moment of $X$,
$S(\nu)$ the fast Fourier transform of $s(t)$, $\nu_{\max}$ the frequency at which $|S(\nu)|$ is maximum, $|S(\nu)|_i$ the $i$th quartile of
$|S(\nu)|$, $DFT(t,\omega)$ is the discrete Fourier transform of $s(t)$, $\omega_2$ the central frequency of $DFT(t,\omega)$, $\omega_{\max}$ the
frequency at the maximum of $DFT(t,\omega)$, $|DFT(t,\omega)|_j$ the $j$th quartile of $|DFT(t,\omega)|$.