# Peer review of "Machine learning prediction of the mass and the velocity of controlled single-block rockfalls from the seismic waves they generate"

_EGUsphere, 2022_

## Referee Comment (RC2)

This manuscript uses data from a controlled rockfall experiment to evaluate relationships between directly measured features of discrete rockfall impacts and the seismic energy generated by these impacts. A conventional approach is initially taken where linear scaling relations between seismic energy or amplitude and kinematic attributes of the impacts are explored. However, the authors also use a supervised machine learning regression to predict rockfall impact parameters (mass and velocity) using >100 features calculated from the seismic data. Labeled data examples (rockfall impacts with measured mass and velocity) are used to train a random forest regressor. Prediction errors of about 11% for velocity and 25% for mass are achieved. The authors examine the importance of each feature in the regression task and find that features associated with frequency-specific seismic energy are particularly influential, consistent with previous work. Operational potential and specific benefits of the machine learning workflow (versus other more conventional methods) are discussed.

Generation of a labeled dataset for supervised machine learning tasks is a significant challenge when working with exotic seismic sources such as rockfalls, as these events are not usually cataloged. The documentation of the rockfall experiment and how the kinematic features of the rock impacts are obtained will be of interest to ESurf readers. The benefits of a machine learning approach for linking seismic observations with desired mass movement parameters (e.g., avoiding the specification of a velocity model, event locations not required) are well-motivated. I appreciate that the authors make a distinction between larger events capable of producing long period seismic energy suitable for inversion versus the smaller events studied here which cannot be analyzed using the same techniques.

The text could benefit from some editing to reduce emphasis on the wavefield attenuation analysis and add more detail elsewhere. I don't think that the introduction and evaluation of the three simple attenuation models is relevant for the thrust of the paper. It is enough to share the "best fit" model and show (via Figs. 3 and 4) the limitations of this. Instead, the authors could further discuss the utility of this work in the context of monitoring. For example, the input to a trained model is presumably the triggered, windowed seismograms for each impact. How might those windowed seismograms be obtained? For the training dataset presented in this work they were manually picked. An application involving the training of a site-specific model for each monitoring location is introduced. Do the authors think that their existing random forest model could instead be generalized to new locations? If not, why not? Presumably this would not be possible for different source types (e.g., granular flow instead of rockfall). The model encompasses both the path/site effects as well as the source physics, though, so generalization to new geologic settings may itself be non-trivial.

In light of the above general comments, I feel this paper requires moderate revisions prior to publication. Please see the specific scientific and technical comments that follow, with line/section/figure numbers provided.

Scientific comments:

| | |
|---|---|
| §2.5 | A figure showing the binary impact time series compared with the seismic signal would be helpful to explain this somewhat unusual process. For example, the binary impact time series could be shown as an additional panel at the top of Fig. 2. This would help readers understand the time lag present as well as how the camera-obtained timings of impacts correspond with the seismic signals of the same. |
| 165, 168 | Eq. 3 is not a linear function of $r$ — though such a linear function is what is plotted as a black line in Fig. 3. Please reconcile this — but also see my synoptic comment on the attenuation model analysis in the third paragraph above. |
| 208–210 | Please provide justification for including the standard deviations of these features, as this is not done in Provost et al. (2017) or Hibert et al. (2017c) — and some of the feature standard deviations are shown to be important in the subsequent feature importance analysis. |
| 252 | This is the first time the mass of the blocks is discussed. Since the random forest predicts mass and velocity for each impact, is the mass expected to be constant? In other words, what is the "real value" used for the mass error analysis? Is it the measured block mass? I assume the modulus of the velocity is taken from the kinematics reconstruction as explained in §2.4? |
| 253 | "real value" — see comment immediately above. Please be explicit about what these values are and how they are obtained. |
| 338–346 | I generally find this paragraph hard to follow. Is "frequency content" referring to a seismic parameter that was predicted like the impact force was in the study cited? Additionally, in the sentence starting on line 342 ("This may suggest…"), what is meant by "a change of the velocity" or "a change of the mass" (for the latter, see also my comment on line 252)? Is the "variability of the impacted forces" being linked with the measured seismic features implicitly in this argument? The final sentence of this paragraph is an important conclusion, but I think the preceding body of the paragraph needs to be rewritten for clarity with definitions in particular made more explicit. |
| 389–390 | The code used for this work is only partially shared in the linked Zenodo repository. The authors should make the code for the rest of the workflow (in particular, the setup/training/testing of the random forest) available in this repository. |

Technical comments:

| | |
|---|---|
| 10 | "and an *a priori* knowledge of the environment" — assumptions about the seismic propagation model are mentioned in the next phrase, so what is this referring to? Please be more specific, or remove. |
| 13 | "constrain" → "constraints on" |
| 16, 18 | Do not capitalize "world" or "earthquake" here. |
| 24 | "complete" seems an odd choice here, since of course there is always more information that can be obtained. Suggest "augment" |
| 42 | "will most of the time generate high-frequency seismic waves" — this energy is always present even if the long-period energy is not prominent. I think "will most of the time" could be removed here. |
| 42–45 | This is an important point that could be rephrased to avoid the false binary of "either small or catastrophic volumes" (what about events in between these two end members?) — suggest something like "...most mass wasting processes, including smaller-volume events, which…" |
| 55 | It isn't surprising that source physics affect the scaling laws, so I suggest that "sometimes even of" be removed here. |
| 63 | For clarity, I suggest noting that the wave propagation model is a challenge for the high-frequency case specifically. |
| 86–94 | I could not find information about the mass and dimensions of the blocks anywhere in the text. This paragraph is a logical place to insert that information. |
| 87–89 | Two methods were used to measure the block shapes. Which method was ultimately chosen? |
| 167 | Note erroneous $p$ inserted before $A(r)$. |
| 191 | Remove "In other words" as what follows are examples (of applications of random forest classification and regression). |
| 196 | I feel the quotes around "predict" can be removed as this terminology is widely used in machine learning (e.g., scikit-learn syntax). |
| Fig. 1 | The color scale used for the trajectories (absolute velocities) should be changed so that it doesn't conflict with the geophone colors and confuse the reader. Also, please include a legend for this color scale (a colorbar). |
| Fig. 2 | Per my comment on §2.5 above, consider adding a panel to the top of this figure which shows the impact time series derived from the camera-based workflow, so that readers can see the correspondence between those impacts and the transients in the seismic waveform. |
| Fig. 3 | See my comment on lines 165, 168 above. The black line does not correspond with Eq. 3. Please reconcile. Also, note that the subpanel letters are not currently referenced in the figure caption. |

| | |
|---|---|
| Fig. 4 | Note that the rainbow color scale is now being used for rockfall launch specification instead of geophone specification, which could confuse readers. Consider using a different color scale. |
| Fig. 6 | It appears that the standard deviation features are plotted with some transparency. Please be explicit about this in the legend or the caption to clarify for the reader. |
| Table A1 | In the table header, "[mean(std)]" is confusing since it appears like a mathematical expression. Perhaps just write "number for standard deviation of feature given in parentheses" or similar. |

---

## Author Comment (AC1)

Dear Editor, Dear Referees,

We thank you for the comments and suggestions you provided to improve the quality and impact of the article we are submitting for publication in E-Surf. In this final response you will find our detailed answers to the questions raised.

We provide answers to each Referee comment below. The answer to a comment is given after repeating the comment and colored in blue.

Best regards,

The Authors

**Referee #1 :**

The manuscript conducted a series of rockfall experiments in the Riou-Bourdoux catchment. The authors utilized the video footage technique to retrieve the rockfall's velocity and seismic record to determine several parameters. Then, the random forest tree algorithm was applied to predict the rockfall's mass and velocity. The authors merged published techniques hitting an excellent point in deciphering two critical parameters, mass, and velocity. However, some basic knowledge of rockfall dynamics, further seismic analysis and other machine learning algorithms should be involved to improve the quality of the manuscript. Thus, I make a decision of major revision to the manuscript.

1.Very detailed background information on seismic techniques for landslide and rockfall research in the introduction. After reviewing the introduction, I supposed that the study is about using a new and accurate reconstruction method of rockfalls' trajectories, then linking rockfalls' dynamic to describe the recorded seismic parameters. However, based on the manuscript title, several machine learning methods can be chosen. Why a Random Forest? What are the pro and coin of a Random Forest? Authors can remove all extraneous information like unnecessary landslide parts, focus on rockfall background review and involve machine learning.

We decided to work with the Random Forests for several reasons. First of all there are the inherent qualities of this machine learning model for classification and regression as demonstrated in previous works (cited in our manuscript). These qualities are the good accuracy generally achieved: in most applications RF outperforms or equals the performance of other algorithms. Secondly the fact that RF is not a black box, you can fully explore the model (the decision trees) visually. Thirdly and most importantly for us, it is possible to test a large number of features without the bad features unduly influencing the prediction result, and it is possible to easily estimate the importance of these features. In our case, as we are as much interested in whether we can predict quantities as in why we can (which features are the most linked to the physical properties), this third quality of the RF algorithm is critical. Finally, RF has been successfully used for many applications to detect and classify signals related to mass-wasting processes, and for operational purposes, one can imagine a future system capable of both detecting, identifying and characterizing slope instabilities using the same RF-based model.

2.Line 69-70. random forest tree is not an innovative approach.

In the context of this study it is more innovative than the classical approaches used to infer the rockfall properties. We can remove the "innovative" if Referee #1 and the editor feels it is important to do so.

3.section 2.1. Rockfalls' physical processes, falling, bouncing, rolling, and sliding associated with the slope angle (Ritchie, 1963). Authors should involve a topographic profile to show the slope angle change along the rockfall trajectory. It is better to put the slope direction map and slope angle map in the supplement material.

All the information about the slope, the blocks and the whole trajectories reconstruction analysis is presented in Noël et al [2022] (link). We will ensure that in the revised paper this is very clear to the reader and refer to the paper by Noel et al. [2022] as a companion paper.

4.Figure 1 lacks the color bar for the translation velocity of rocks.

We will add this information in the revised manuscript

5.section 2.2. Rocks' geometry is a vital parameter for rockfall trajectory (Caviezel et al., 2021). The authors should put rocks' photos and dimensions in supplement material.

See comment above on the companion paper by Noel et al. [2022]

6.Line 101-102. Where are the four locations with a terrestrial laser scanning device? Please mark them on the map.

See comment above on the companion paper by Noel et al. [2022]

7.Can authors offer one video for a rockfall experiment to help the reader understand the whole process?

See comment above on the companion paper by Noel et al. [2022]

8.Line 131. Falling is a mechanism term of rockfall linking to the slope angle. Remove free-falling.

We will revise the manuscript according to this suggestion.

9.Line 158-159. What are the exact distances between the source and the receivers? Is that point-to-point distance or the topographic distance? Both distances do not affect the big picture when the source and the receivers are close. However, gradually enlarging the source-to-receiver distance and the seismic wave transmits through high-relief topography. Two different distances may produce the error for the body and surface wave model. The DEM did not cover the GEO1 to GEO8. So, authors can use GEO9 to GEO16 and the early to middle stage of experiments to explain the effect, then put test results in supplement material.

We used the 3D point to point distance without considering the topography. We will add this clarification in the revised manuscript. In an early stage of this work presented at EGU we did a analysis which is kin to what Referee #1 is asking, we think. You can find an excerpt of the poster

below. This figure and the table shows that there is no effect of the distance on the best fit of the amplitude as a function of the distance. The best fit is always observed for the body-wave model.

**Block #1 propagation :** The block #1 was the block instrumented with the accelerometer /gyroscope positioned close to its centre-of-mass. Its mass was 125 kg. After 8 impacts the block broke in half at the 9th impact. The two fragments continued to propagate along the path. The timing of the impacts of each of the two fragments were identified thanks to the videos.

[Figure]

**Fig. 3:** Left panel : Seismic signals recorded at each geophone. Right panel: Maximum envelope amplitude as a function of the distance and smoother FFT spectrum for each impact and each station. The colours corresponds to the colour of the geophones on Fig. 1.

**Amplitude attenuation :** Fig. 3 shows the attenuation of the amplitude of seismic waves generated by each impact, i.e. the amplitude of the signal is decreasing with the distance of the sensor to the location of the impact. Thanks to the trajectory reconstruction we can determine the distance between each impact and each geophone. Then we can compute attenuation models and find the one that better explain the observed decay of amplitude with distance. Determining an adequate attenuation model is critical to determine the parameters (amplitude and energy) at the source which are then compared to dynamic parameters.

We tested two simple models, one for surface wave (Eq. 1) and one for body wave (Eq. 2), as well as the possibility of a simple linear attenuation of the amplitude with distance (Eq. 3). For each model we computed the regression line and assess the quality of the regression by computing $R^2$ (Tab. 1).

| Impact # | Model 1 | Model 2 | Model 3 |
|---|---|---|---|
| 1 | 0,91 | 0,89 | 0,88 |
| 2 | 0,88 | 0,85 | 0,85 |
| 3 | 0,84 | 0,79 | 0,76 |
| 4 | 0,89 | 0,86 | 0,82 |
| 5 | 0,71 | 0,69 | 0,48 |
| 6 | 0,69 | 0,61 | 0,39 |
| 7 | 0,91 | 0,88 | 0,86 |
| 8 | 0,60 | 0,47 | 0,29 |
| 9 | 0,77 | 0,71 | 0,59 |
| 10 | 0,62 | 0,52 | 0,21 |
| 11 | 0,75 | 0,69 | 0,34 |
| 12 | 0,33 | 0,21 | 0,03 |

**Tab. 1:** $R^2$ from the regression with each model of the distance and amplitude data for each impact

**Model 1 : Body Waves**

$$A(r) = A0 \frac{e^{-\beta r}}{r} \quad (Eq.1)$$

**Model 2 : Surface Waves**

$$A(r) = A0 \frac{e^{-\beta r}}{\sqrt{r}} \quad (Eq.2)$$

**Model 3 : Simple Linear att.**

$$A(r) = \frac{A0}{r} \quad (Eq.3)$$

Eq. 1, 2 and 3 : A0 is the amplitude at the source, r the distance to the source and β the anelastic attenuation factor.

10.section 2.7. The authors cited several references to support the method of Random Forests, but it could be more explicit. What are the criteria, gini or entropy, in this research? Does the author set max depth for each tree? How to generate a result of OOB score in Figure 6? Random Forests is a black box to produce the result whose opaque process means that implementers must fully trust the model result and cannot understand details.

We used the default parameters of the TreeBagger function of the matlab machine learning toolbox, which are the same as the RandomForest function from other toolboxes (e.g. scikit-learn in Python). The only parameter we set was the number of trees. Nevertheless we will add the details requested by Referee #1 in the revised manuscript. The OOB score was computed by permutation of the features values. We will also add this information in the manuscript.

Other machining learning techniques like XGBoost, cluster model …etc. needed to be considered to double-check the result of Random Forests. Further, EGU society encourages authors to offer the source code to allow other researchers reproduce the result. I suggested that the authors release the machining learning code when the paper is published.

We are not convinced that other machine learning techniques are needed at this stage of this analysis. In this paper we wanted to show that a machine learning algorithm can be used to perform the task of retrieving the mass and the velocity of rockfalls. We choose the RF for the reasons aforementioned. We do not want to benchmark machine learning techniques to perform this task yet. As for the use of cluster models we are curious how those would help and what kind of model Referee #1 would suggest to perform the same task. We use an off-the-shelf implementation of the RF algorithm. We used the one provided by the Machine Learning toolbox from Matlab because we had former codes to compute the scaling laws in this language. However this toolbox is very expensive and we think it would be wrong to encourage readers to use this toolbox when free and equally powerful versions exist in Python for example, which is not a copyrighted language. The whole analysis can easily be reproduced in Python with the data and the information we provided. Nevertheless we will do our possible to provide a Python version of the machine learning codes once the paper is published, as suggested by Referee #1.

11.section 3.1. What $\beta$ values are in the rockfall experiment? When adapting the body wave model, those $\beta$ values are located in the ideal range or not.

We do not understand what Referee #1 is referring to when mentioning "ideal range". The $\beta$ values are in the same range as the one obtained in a previous experiment [Hibert et al., 2015].

12.Line 238-239. "Low $R^2$ values might be explained by irregular kinematic behaviors such as the block hitting an obstacle (trees, other rocks) " Also, sliding shows typically in the early stage of a rockfall, with significant impact after energy loss or near the stopping moment. Please offer a video or time-lapse photos to support the result.

There was no sliding in the early stage of those triggered rockfalls. For videos see comment above on the companion paper by Noël et al. [2022]

13.The dark and light grey lines in Figure 3 are hard to see.

We will improve this figure in the revised version.

14.section 3.3. what is the distribution of predicted values? Underestimate or overestimate the actual values (velocity/ mass)? An extra figure for predicted values should be included.

We will add histograms showing those differences in the revised version of the paper.

15.section 3.4. Lengthy in this section. In Figure 6(a), #13 gets a more substantial OBB score than others and the result. In Figure 6(b), the OBB score of the top 6-10 is close to the top 4-5. Only describing the top 3 parameters of velocity and mass result is enough where their OBB scores are higher than 0.3.

We feel there is information and discussion to have on more than the first three best features so we prefer to keep this section like it is in the original version of the paper.

16. Line 285-287. How about the kinetic energy after impact? It is no difference in $R^2$ between 0.39 and 0.34(Figure 4g, 4h). Figure 4 should include a confidence interval around the slope of a regression line.

Given the low R2 we believe that showing a confidence interval around the slope of the regression line might mask the whole plot but we will try this suggestion and modify the figure according to those tests.

17. Line 287-289. Although the rock boulder moved in the West-East direction, the seismic wave transmitted from South to North in the initial stage of rockfall and West-East in the middle stage. Different stages may present different features. Putting different stages of data in one figure is unfair. Further, the poorest correlation observed between Viy, and $A_0$ may be caused by the signals near GEO1 and GEO16, which is worth examining the signals from which experiments and what stage of rockfall bring the poorest correlation.

The general direction of the rockfall is always the same. There are no differences in the initial and the middle stage (see videos in Noëll et al. [2022]).

18. Line 308. what is the definition of spectrum width?

In this case it would be the variance of the FFT spectra. We will clarify this in the revised version.

19. Line 323-324. "This suggests that the difference of seismic energies recorded at different stations is important information for the prediction of the velocity (not for the mass)." Why not use signals from cluster stations to execute Random Forest and see their difference? For example, GEO1 to GEO3 and GEO11 to GEO 13 can be grouped. Stations in each group inherit similar paths and site effects, which can reveal how distance affects the results.

The very purpose of our study, and the results we produce, show that the propagation distance does not affect the RF result. This is the interest of this approach, to be able to determine the mass and velocity without the need to localize the impacts.

20. Line 324-325. Huang et al. (2007) already conducted drop experiments of individual rocks, showing that the larger stones generate the extending feature of lower frequency signals.

Thanks for these suggestions, we will add the references in the manuscript.

21.The cause and effect should be clarified between Lines 326-331.

We will clarify this part in the revised manuscript.

22. Parameters of ES1 to ES5 are crucial to mass and velocity prediction. Let the y-axis of figure2b be a log scale, which highlights the low-frequency band.

A log scale will deform the spectrogram in the high-frequency band. We used only frequencies above 1Hz in our features bank. We think that a linear scale here is better for the readers to have the best readability of the information given by the spectrogram.

23. Perspective: What are the installation criteria, like station geometry, for further research? Is it necessary to have ten stations with linear arrays? Is all stations equally important for machine learning? If not, what is the restriction? Can further research transfer learning from this research? After reviewing the manuscript, I supposed the authors should answer those questions rather than extraneous information in section 5 to enhance the quality of the paper.

Those are indeed very interesting questions but which are a step further of what we propose here. In this study we want to show that machine learning can perform the task of predicting the mass and the velocity of rockfalls without having to localize them. We also wanted to show that the ML approach performs better in this case than those based on scaling laws. Future work will allow us to reproduce the same experiment in other contexts, to work on natural events, and to assess the transferability of models, but with the data we present here we cannot yet answer those questions.

**Referee # 2 :**

17 February 2023 This manuscript uses data from a controlled rockfall experiment to evaluate relationships between directly measured features of discrete rockfall impacts and the seismic energy generated by these impacts. A conventional approach is initially taken where linear scaling relations between seismic energy or amplitude and kinematic attributes of the impacts are explored. However, the authors also use a supervised machine learning regression to predict rockfall impact parameters (mass and velocity) using >100 features calculated from the seismic data. Labeled data examples (rockfall impacts with measured mass and velocity) are used to train a random forest regressor. Prediction errors of about 11% for velocity and 25% for mass are achieved. The authors examine the importance of each feature in the regression task and find that features associated with frequency-specific seismic energy are particularly influential, consistent with previous work. Operational potential and specific benefits of the machine learning workflow (versus other more conventional methods) are discussed.

Generation of a labeled dataset for supervised machine learning tasks is a significant challenge when working with exotic seismic sources such as rockfalls, as these events are not usually cataloged. The

documentation of the rockfall experiment and how the kinematic features of the rock impacts are obtained will be of interest to ESurf readers. The benefits of a machine learning approach for linking seismic observations with desired mass movement parameters (e.g., avoiding the specification of a velocity model, event locations not required) are well-motivated. I appreciate that the authors make a distinction between larger events capable of producing long period seismic energy suitable for inversion versus the smaller events studied here which cannot be analyzed using the same techniques.

The text could benefit from some editing to reduce emphasis on the wavefield attenuation analysis and add more detail elsewhere. I don't think that the introduction and evaluation of the three simple attenuation models is r elevant for the thrust of the paper. It is enough to share the "best fit" model and show (via Figs. 3 and 4) the limitations of this.

We fully understand this remark by Referee #2. We asked ourselves the same question. What pushed us to show the three models is that in the literature it is almost accepted that the attenuation model using surface waves was the one to choose. However, our study shows that it is more complex than that, and that at least in the near field, a model based on volume waves is more relevant. Even if this result is not directly related to the rest of our analysis, we think that it is nevertheless important to communicate it to the community interested in seismic signals generated by rockfalls. However, it is true that the third model involving attenuation only with distance adds nothing to this discussion and we therefore remove it from the manuscript and Figure 3 as suggested.

Instead, the authors could further discuss the utility of this work in the context of monitoring. For example, the input to a trained model is presumably the triggered, windowed seismograms for each impact. How might those windowed seismograms be obtained?

This is the object of other research we do for the detection and the classification of rockfall seismic signals using the same machine learning algorithm (Random Forests). The aim of all these researches is to eventually be able to propose a system for detecting, classifying and characterizing the properties of rockfalls that would integrate all machine learning approaches for near real time monitoring. We are still far from being able to propose such a system, but all our recent works, including this study, is contributing to it.

For the training dataset presented in this work they were manually picked. An application involving the training of a site-specific model for each monitoring location is introduced. Do the authors think that their existing random forest model could instead be generalized to new locations? If not, why not? Presumably this would not be possible for different source types (e.g., granular flow instead of rockfall). The model encompasses both the path/site effects as well as the source physics, though, so generalization to new geologic settings may itself be non-trivial.

Yes it would be non-trivial and we cannot answer this question with only this analysis and yes, the trained model for this specific context might not be transferable in other contexts. This is a point that we underline in our conclusion. This first study that we are proposing is more intended to serve as a "proof of concept", and to demonstrate that the new approach we are proposing is efficient. The question of the transferability of models will arise in a second step, when we will have access to data

from other contexts acquired through other experiments. It is a more global challenge in seismology and geophysics in general, to have fairly large datasets that would allow to have a model trained for all possible contexts. It is not yet the case, but we hope that this study and others will demonstrate the relevance of machine learning approaches for the study of environmental processes, and push our community to develop shared databases. However, the effort required to build these databases can only be justified if we can demonstrate the relevance of these approaches. We believe that the study we are proposing participates in it.

While the transferability of the machine learning model trained with our experiment might be complex, the transferability of the approach would however be relatively easy. In our study we deployed a large array of sensors to get a maximum of accurate information from the experiment on both the dynamics of the blocks and their seismic signals. However, if someone wants to deploy this kind of approach for an operational application, they would only have to deploy their seismic sensors network and throw 10 to 30 blocks within their network to get enough data to train a model to be able to predict the mass of the blocks. If they want to predict the velocity then it would need a bit more field work, such as using a mobile GNSS to find the positions of each impact for each block and then compute the impact velocities. Other approaches, based on physical models, would require the same work (especially to calibrate the scaling laws), but would require to have a good attenuation model of the medium (through seismic tomography) and a method to accurately localize each impact for each new event, all while having a probably worse accuracy. One of the strengths of the RF approach is that we do not need an attenuation model and we do not need to localize the impact to be able to get an estimate of the block mass and velocity.

In light of the above general comments, I feel this paper requires moderate revisions prior to publication.
Please see the specific scientific and technical comments that follow, with line/section/figure numbers provided.

Scientific comments:

§2.5 A figure showing the binary impact time series compared with the seismic signal would be helpful to explain this somewhat unusual process. For example, the binary impact time series could be shown as an additional panel at the top of Fig. 2. This would help readers understand the time lag present as well as how the camera-obtained timings of impacts correspond with the seismic signals of the same.

This is a great suggestion and we will modify figure 2 accordingly.

165, 168 Eq. 3 is not a linear function of r — though such a linear function is what is plotted as a black line in Fig. 3. Please reconcile this — but also see my synoptic comment on the attenuation model analysis in the third paragraph above.

See our previous response to your comment on the attenuation model.

208–210 Please provide justification for including the standard deviations of these features, as this is not done in Provost et al. (2017) or Hibert et al. (2017c) — and some of the feature standard deviations are shown to be important in the subsequent feature importance analysis.

In contrast to the studies mentioned, here we aggregate the features of several seismic signals within a single event (in the ML term). To do this, we average the feature values of the seismic signals recorded at all the stations for the same impact. In addition to this average, in order to include the dispersion of these values for the same impact, we have chosen to also include the standard deviation as a feature. We will reformulate and detail this choice in the revised version.

252 This is the first time the mass of the blocks is discussed. Since the random forest predicts mass and velocity for each impact, is the mass expected to be constant? In other words, what is the "real value" used for the mass error analysis? Is it the measured block mass? I assume the modulus of the velocity is taken from the kinematics reconstruction as explained in §2.4?

We removed from our dataset every impact that resulted in a breaking of the block. We have kept only the impacts for which the block did not undergo major changes according to our visual observations. We cannot exclude a marginal change in the mass of the block due to successive impacts, but this should not have a major impact on our results.

253 "real value" — see comment immediately above. Please be explicit about what these values are and how they are obtained.

We will add the requested clarification in the revised manuscript.

338–346 I generally find this paragraph hard to follow. Is "frequency content" referring to a seismic parameter that was predicted like the impact force was in the study cited? Additionally, in the sentence starting on line 342 ("This may suggest…"), what is meant by "a change of the velocity" or "a change of the mass" (for the latter, see also my comment on line 252)? Is the "variability of the impacted forces" being linked with the measured seismic features implicitly in this argument? The final sentence of this paragraph is an important conclusion, but I think the preceding body of the paragraph needs to be rewritten for clarity with definitions in particular made more explicit.

We will rewrite this entire paragraph to clarify our argument and our thinking.

389–390 The code used for this work is only partially shared in the linked Zenodo repository. The authors should make the code for the rest of the workflow (in particular, the setup/training/testing of the random forest) available in this repository.

See our response to the similar comment made by referee #1

Technical comments:

We are very thankful for all these insightful technical comments from Referee #2. We will include all of them in the revised version.

"and an a priori knowledge of the environment" — assumptions about the seismic propagation model are mentioned in the next phrase, so what is this referring to? Please be more specific, or remove.

13 "constrain" → "constraints on"

16, 18 Do not capitalize "world" or "earthquake" here.

24 "complete" seems an odd choice here, since of course there is always more information that can be obtained. Suggest "augment"

42 "will most of the time generate high-frequency seismic waves" — this energy is always present even if the long-period energy is not prominent. I think "will most of the time" could be removed here.

42–45 This is an important point that could be rephrased to avoid the false binary of "either small or catastrophic volumes" (what about events in between these two end members?) — suggest something like "...most mass wasting processes, including smaller-volume events, which…"

55 It isn't surprising that source physics affect the scaling laws, so I suggest that "sometimes even of" be removed here. 63 For clarity, I suggest noting that the wave propagation model is a challenge for the high-frequency case specifically.

86–94 I could not find information about the mass and dimensions of the blocks anywhere in the text. This paragraph is a logical place to insert that information.

87–89 Two methods were used to measure the block shapes. Which method was ultimately chosen?

167 Note erroneous p inserted before A(r).

191 Remove "In other words" as what follows are examples (of applications of random forest classification and regression).

196 I feel the quotes around "predict" can be removed as this terminology is widely used in machine learning (e.g., scikit-learn syntax).

Fig. 1 The color scale used for the trajectories (absolute velocities) should be changed so that it doesn't conflict with the geophone colors and confuse the reader. Also, please include a legend for this color scale (a colorbar).

Fig. 2 Per my comment on §2.5 above, consider adding a panel to the top of this figure which shows the impact time series derived from the camera-based workflow, so that readers can see the correspondence between those impacts and the transients in the seismic waveform.

Fig. 3 See my comment on lines 165, 168 above. The black line does not correspond with Eq. 3. Please reconcile. Also, note that the subpanel letters are not currently referenced in the figure caption. 3

Fig. 4 Note that the rainbow color scale is now being used for rockfall launch specification instead of geophone specification, which could confuse readers. Consider using a different color scale.
Fig. 6 It appears that the standard deviation features are plotted with some transparency. Please be explicit about this in the legend or the caption to clarify for the reader.

Table A1 In the table header, "[mean(std)]" is confusing since it appears like a mathematical expression. Perhaps just write "number for standard deviation of feature given in parentheses" or similar. 4

---

## Author Response (AR3)

**Clément Hibert**
Associate Professor

hibert@unistra.fr

**Institut Terre et
Environnement de
Strasbourg (ITES)
- CNRS UMR 7063**
5 rue René Descartes
F-67084 Strasbourg Cedex

Strasbourg, 25 March 2024

Object : Response to technical corrections

Dear Prof. Dr. Michael Krautblatter,

We revised the manuscript by following all your suggestions. We added supplementary material with the figure and the corresponding discussion from the EGU poster. We included the elements of discussion in the revised paper as suggested. We supplemented the acknowledgement with thanks toward the reviewers and the editors. The only comment for which we did not follow your suggestion regards the position of the four location of the TLS on the map : Given that the measurement points are distant from the DTM shown in Figure 1, providing the positions of the TLS to scale would significantly reduce the size and thus the readability of the map presented in Figure 1.

We hope this version is now suitable for publication in E-Surf, and we thank you warmly for all your help and constructive suggestions.

Best Regards,

Clément Hibert, on behalf of all the authors